# Mechanism of Delayed Convulsion in Fish: The Actions of Norepinephrine in Spinal Cord

**Cheng-Linn Lee, Yuri Kominami and Hideki Ushio \***

Graduate School of Agricultural and Life Sciences, University of Tokyo, 1-1-1 Yayoi, Tokyo 113-8657, Japan; chenglinnri@g.ecc.u-tokyo.ac.jp (C.-L.L.); akomi@mail.ecc.u-tokyo.ac.jp (Y.K.)

\* Correspondence: aushio@mail.ecc.u-tokyo.ac.jp; Tel.: +81-3-5841-5300

**Abstract:** Cranial spiking (CS) is among the most popular slaughtering methods for delaying the rigor mortis progress of fish muscles. However, it may cause a convulsion (subsequently referred to as delayed convulsion), which undermines the meat quality and taste. This study aimed to elucidate the mechanism underlying the delayed convulsion and examine its influence on ATP consumption. Ten carps, nine tilapias, ten rainbow trouts, two ayus, three greenling, thirty-five red seabreams, two striped jack and two stone flounders underwent CS around the medulla oblongata area, which induced different delayed convulsion profiles specific to each species. To investigate the norepinephrine (NE) actions related to delayed convulsion, 27 red seabreams, a representative fish species that exhibits delayed convulsion, were treated with a monoamine-depleting agent, reserpine, or with a monoamine oxidase inhibitor, pargyline, two hours before CS. Spinal cord destruction (SCD) was employed to completely prevent spinal cord functions of the fish in another group. Compared with the control group (CS only), the reserpine, pargyline, and SCD groups showed significantly inhibited delayed convulsion and ATP consumption. This suggests that delayed convulsion is the main ATP-consuming response. Our findings suggest that delayed clonic convulsion in red seabreams is associated with the rapid decrease in spinal cord NE levels, which triggered the rebound motor neuron hyperactivity.

**Keywords:** spinal reflex; clonic convulsion; red seabream; norepinephrine

## 1. Introduction

*Sashimi* is the Japanese term used for raw meat of fishes and shellfishes that have gained worldwide recognition [1,2]. With this rapid rise in popularity, raw fish consumption has significantly increased worldwide. *Sashimi* freshness is the most important factor for consumption together with taste, flavor, and meat texture. Because the property greatly depends on the conditions of fish muscle cells after sacrificed, the suppression of physiological stresses such as struggling is important for preserving *Sashimi* freshness. Numerous techniques are currently used to sacrifice the fish. For example, in Japan, cranial spiking (CS) around the medulla oblongata (hindbrain region) is among the most widely adopted methods and is known as efficient with respect to freshness preservation and meat texture [3,4]. However, this procedure can cause a drastic convulsion at 2–20 postoperative minutes in several fish species, such as Kahawai (*Arripis trutta*) [5] and horse mackerel (*Trachurus japonicus*) [6]. Although this convulsion causes rapid ATP consumption in muscle, which could lead to deterioration of the fish freshness and flesh texture [7], the physiological mechanisms for the convulsion are still ambiguous. The post-decapitation reflex (PDR) in mammals also induces a similar spontaneous ATP-consumption. PDR is presented by rodents and several other animals except for guinea pigs [8,9]. Specifically, in rodents, decapitation leads to a few seconds of body quiescence, followed by a short-lasting convulsion for up to 20 secs. In livestock, pithing has been shown to cause immediate tonic convulsion, followed by clonic convulsions lasting for up to a few minutes [10,11].

PDR could critically involve noradrenergic fibers originating from the supraspinal region [12]. Moreover, selective norepinephrine (NE) depletion in the spinal cord in rats through intraperitoneal 6-OHDA administration significantly inhibited PDR [13]. Fukuda et al. [14] reported dose-dependent inhibition of decapitation- and electroshock-induced clonic convulsion by 6-OHDA, which selectively destroyed catecholaminergic terminals in the brain. Blocking and stimulating postsynaptic and presynaptic spinal cord $\alpha_1$-adrenoceptors, respectively, can inhibit PDR [15]. This suggests the involvement of bulbospinal norepinephrinergic fibers in clonic convulsion movements.

Endogenous norepinephrine (NE) is widely known for its anticonvulsant activity in the treatment of refractory epilepsy [16,17]. NE system deficiency is implicated in increased susceptibility to seizure-inducing stimuli, including sound and drugs [18]. Moreover, catecholamines and 5-hydroxytryptamine (5-HT), which have less potency, can partially induce skeletal muscle hyperpolarization in mice via nonspecific hormone-membrane interaction [19]. This accumulating evidence suggests that NE can modulate skeletal muscle actions through norepinephrinergic innervation and the endocrine system (Figure 1).

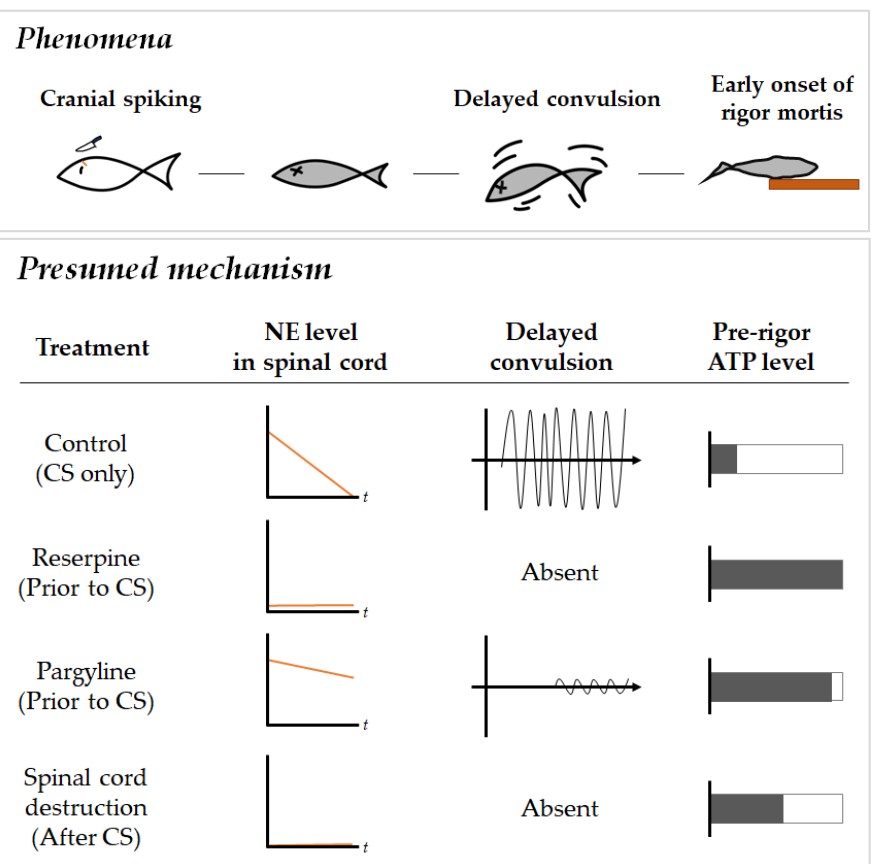

**Figure 1.** Diagram of delayed convulsion phenomenon and proposed mechanism [20]. Rate of NE depletion plays an important role in the delayed convulsion pattern which affects the pre-rigor ATP levels.

Although mammalian PDR and post- cranial spiking (CS) delayed convulsion in fish are involuntary motions observed after decapitation, their time course profiles significantly differ from each other. The goal of the present study is to disclose the physiological mechanisms underlying the delayed convulsion in fish.

## 2. Results

### 2.1. Among-Species Differences in Delayed Convulsion Measures

We examined the latency and magnitude of delayed convulsion in fishes that were cultivated in freshwater or saltwater environments as follows: freshwater (carp *Cyprinus carpio*, tilapia *Oreochromis niloticus*, ayu *Plecoglossus altivelis* and rainbow trout *Oncorhynchus mykiss*) and saltwater (greenling *Hexagrammos otakii*, red seabream *Pagrus major*, striped jack *Caranx delicatissimus*, and stone flounder *Kareius bicoloratus*). Table 1 shows an overview of delayed convulsion profiles of different fish species. Saltwater fish species tended to exhibit delayed convulsion with higher intensity and longer duration. Among these freshwater fish species, carp and tilapia did not present delayed convulsion. Ayu and rainbow trout, instead, exhibited mild delayed convulsion. The average onset time and duration of delayed convulsion in the freshwater fishes that convulsed were approximately 3 and 1 min, respectively. Saltwater fishes, except the greenling, showed the most violent convulsions. The latency time in the red seabream and striped jack was approximately 20 min. Notably, the latency time in stone flounder lasted for up to one hour. The average convulsion duration was 15, 30, and 40 min in the red seabream, striped jack, and stone flounder, respectively. Moreover, fishes with larger sizes tended to exhibit stronger convulsion patterns.

**Table 1.** Delayed convulsion profiles of different fish species. Delayed convulsion components were registered immediately after sacrifice by CS.

| Fish Species (Individual Numbers) | Onset Time (min) | Duration (min) | Magnitude |
|---|---|---|---|
| Freshwater | | | |
| Carp *Cyprinus carpio* (10) | — | — | — |
| Tilapia *Oreochromis niloticus* (9) | — | — | — |
| Rainbow trout *Oncorhynchus mykiss* (10) | ~3 | ~1 | + |
| Ayu *Plecoglossus altivelis* (2) | ~3 | ~1 | + |
| Saltwater | | | |
| Greenling (Ainame) *Hexagrammos otakii* (3) | ~7 | ~5 | + |
| Red seabream *Pagrus major* (35) | ~20 | ~15 | +++ |
| Striped jack (Shima-aji) *Caranx delicatissimus* (2) | ~20 | ~30 | +++ |
| Stone flounder (Ishigarei) *Kareius bicoloratus* (2) | ~60 | ~40 | +++ |

### 2.2. Effects of Pharmacological Compounds Involving Monoamine Metabolism on Delayed Convulsion in Red Seabreams

The control group showed delayed convulsions 5–25 min after sacrifice, with a convulsion magnitude of $396 \pm 223$ gw (Figure 2A). The reserpine group showed small convulsions until 25 min after sacrifice. Notably, two fish in the reserpine group presented weak and spontaneous convulsions in the swimming tank before sacrifice. Two and all fish in the pargyline and spinal cord destruction (SCD) groups, respectively, showed no convulsion. Pretreatment with pargyline decreased the convulsion magnitude; moreover, it prolonged the latency rather than the duration of delayed convulsion (Figures 1C and 2B).

### 2.3. Relationship between Delayed Convulsion and Changes in ATP and Creatine Phosphate Levels

ATP used during muscle contraction is rapidly recovered using a creatine kinase system that converts ADP and creatine phosphate (CP) into ATP and creatine [21]. Given the insignificant lactate accumulation in all fast muscles of fish (data not shown), we estimated the ATP equivalent level as a sum of ATP and CP levels. The initial ATP equivalent level immediately after CS was $16.9 \pm 1.82$ μmol/g muscle (the initial group). After the convulsions had subsided (around 25 min post-sacrifice), ATP equivalent levels decreased to $2.20 \pm 1.47$, $5.66 \pm 1.76$, $17.2 \pm 1.42$, and $14.9 \pm 3.4$ μmol/g for the control, SCD, reserpine, and pargyline groups, respectively. As shown in Figure 3, the ATP equivalent levels were negatively correlated with the convulsion magnitude levels.

### 2.4. Changes in Spinal Cord Norepinephrine and Monoamine Levels

The spinal cord NE levels of the control, reserpine, and pargyline groups were $1.39 \pm 0.3$, $0.47 \pm 0.28$, $1.68 \pm 0.54$ nmol/g wet tissue, respectively, immediately after death (Figure 4A). In the control group, the NE level decreased to $0.85 \pm 0.38$ and $0.24 \pm 0.23$ nmol/g wet tissue after 5 and 20 min, respectively. Contrastingly, the reserpine group showed no significant decrease in the NE level between the pre- and post-convulsion states. The pargyline group exhibited a slight decrease to $1.35 \pm 0.63$ nmol/g and $1.05 \pm 0.49$ wet tissue in the pre-convulsion and convulsion states, respectively. Although only DA was detectable among monoamines, the level of DA was very low, 0.05–0.14 nmol/g wet tissue in the control group, and was only found significantly changed until the post-convulsion state (Figure 4B).

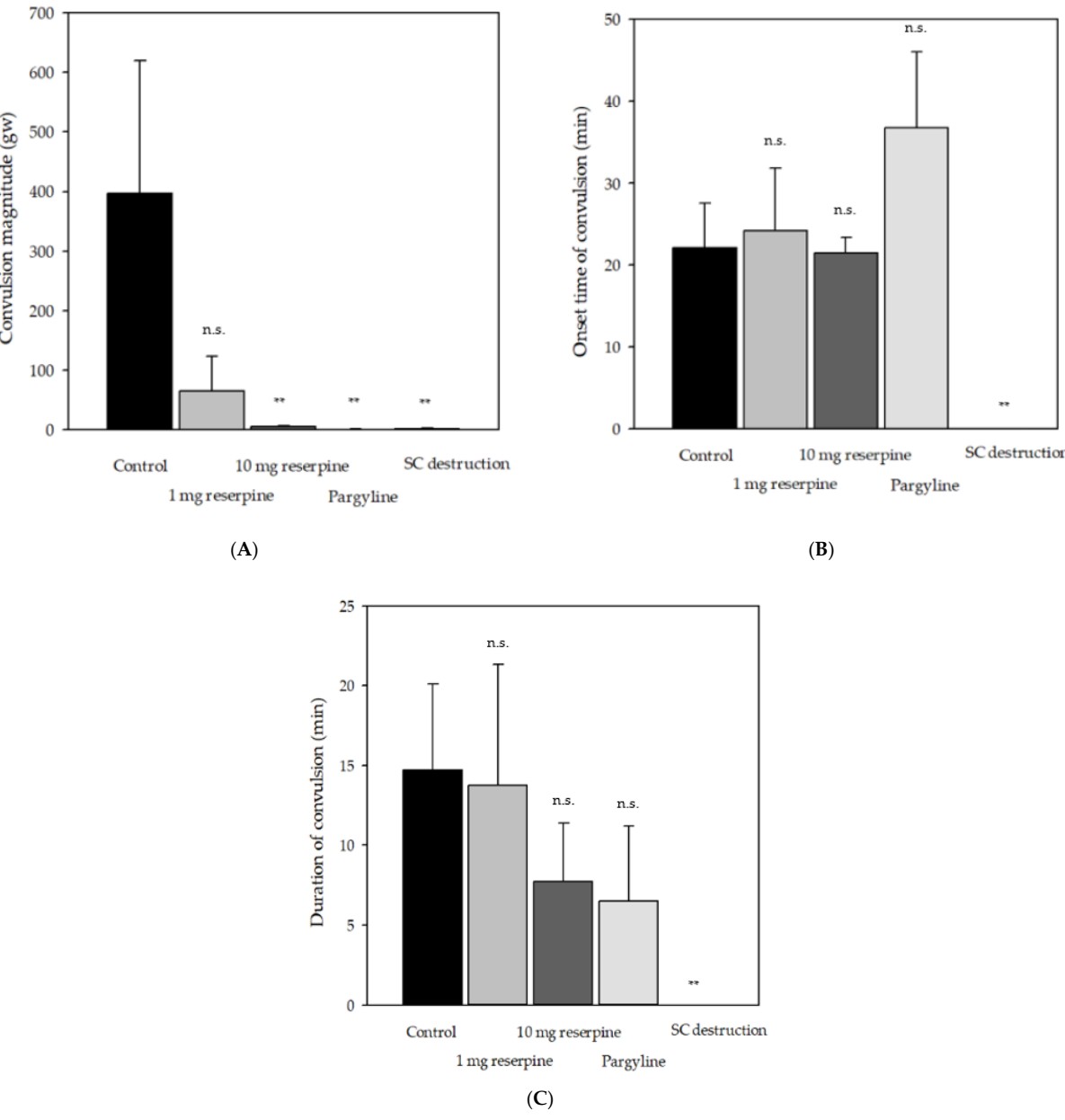

(A)

(B)

(C)

**Figure 2.** The comparison of delayed convulsion patterns of red seabreams subjected to indicative treatments: (**A**) the magnitude of convulsion, (**B**) the latency time of convulsion, and (**C**) the duration of convulsion. Indicative treatments were applied to the red seabreams (*Pagrus major*) before (reserpine or pargyline) or after spinal cord destruction (SCD) cranial spiking (CS). Values represent mean $\pm$ SD of nine individuals. Statistically significant difference is shown as * $p < 0.05$, ** $p < 0.01$. Not significant, n.s.

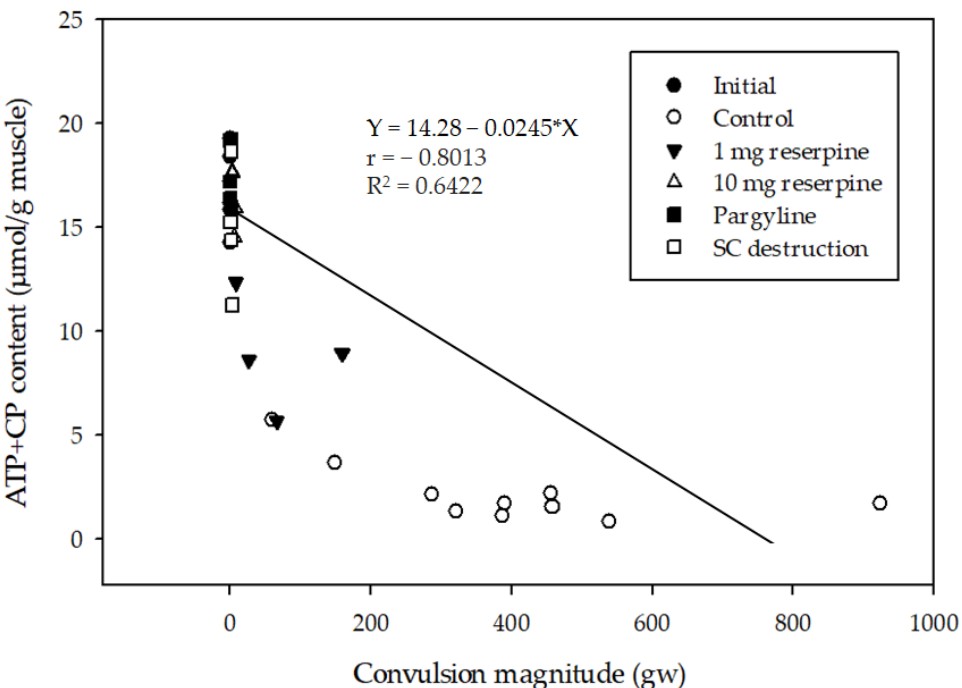

**Figure 3.** Correlation between the magnitudes of delayed convulsion and remaining ATP + creatine phosphate (CP) contents in the dorsal skeletal muscles at the post-convulsion state. Fish subjected to different treatments are represented as: ○ (control), ▼ (1mg reserpine), △ (10 mg reserpine), ■ (pargyline) and □ (spinal cord destruction). ● represents the initial amount of ATP equivalent in pre-convulsion state. Simple linear regression relates X to Y through an equation of $Y = 14.28 - 0.0245 \times X$, $r = -0.8013$. r: correlation coefficient, $R^2$: coefficient of determination.

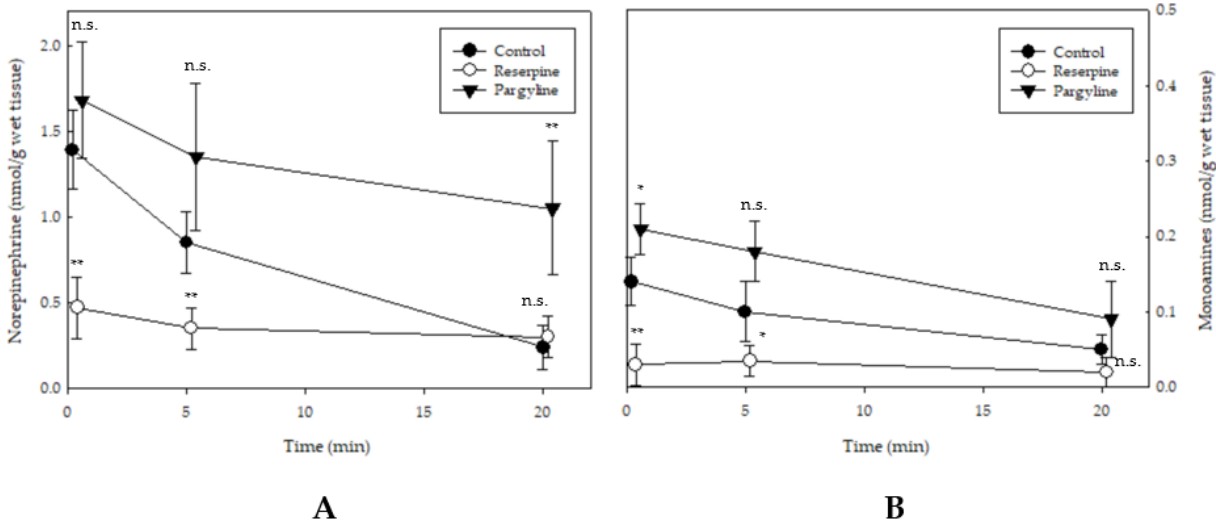

**A** **B**

**Figure 4.** Variation of norepinephrine (**A**) and other monoamine levels (dopamine, **B**) in the spinal cords of red seabreams administrated with indicative pharmacological inhibitors. Fish subjected to different treatments are represented as: ● (control), ⊖ (10 mg reserpine), and ▼ (pargyline). Statistically significant difference is shown as * $p < 0.05$, ** $p < 0.01$. Not significant, n.s.

### 3. Discussion

In this study, we examined delayed convulsion characteristics in several fish species and investigated monoaminergic involvement in delayed convulsion. We observed delayed convulsions in several fish species, including ayu, rainbow trout, greenling, red seabream, striped jack, and stone flounder, but not in carp and tilapia. The red seabream specifically

exhibited a representative delayed convulsion pattern that showed moderate latency and duration. Our findings indicated that saltwater fish species presented more often, stronger, and longer delayed convulsions. Contrastingly, the freshwater fish species carp and tilapia did not present convulsive responses, and ayu and rainbow trout exhibited relatively weaker and shorter convulsion. It is of interest to know how convulsion patterns associate with these fish species. Although it appears that the convulsion patterns are related to the salinity of habitats which may influence the plasma ion levels, we cannot attribute the convulsion occurrence to higher musculature salt contents with the limited information available.

In this study, the onset time of delayed convulsion varied from ~3 to ~60 min, depending on the fish species. Contrastingly, PDR in decapitated rodents usually occurs with a duration of about 20 s after a few seconds in the quiescent state [22]. The occurrence of violent convulsions or involuntary muscle contractions after eliminating descending influences is ubiquitous in numerous vertebrates, including PDR in rats [23], rostral wiping reflex in decapitated frogs [24,25], and the spontaneous autotomized gecko tail movement [26]. Compared with mammals, poikilotherms present a much longer quiescent state, which resembles the spine shock phenomenon in acute spinal animals. The presence or absence of delay in post-decapitation clonic convulsion is an important difference between mammal PDR and delayed convulsions in fish; however, they may have common post-decapitation nervous processes. Roberts et al. [13] hypothesized several PDR causes, including decapitation-induced mechanical stimulation of descending noradrenergic systems, loss of tonic supraspinal inhibition of spinal reflexes, and activation of the central pattern generator (CPG) by NE released upon spinal cord transection. These hypotheses are indicative of norepinephrinergic system involvement in the delayed convulsion.

Based on the aforementioned results, we selected the red seabream as a model organism to further investigate the neural mechanisms underlying delayed convulsions in fish. In the red seabream, reserpine injections showed dose-dependent suppression of convulsive movements; while, pargyline prolonged the latency time, respectively (Figure 2). Pretreatment of red seabreams with 1 mg reserpine 2 h before CS significantly prevented the convulsion phenomenon; however, it did not prolong the latency time (Figures 1B and 2A). Reserpine is an efficient monoamine-depleting agent [27]. It inhibits NE and 5-HT reuptake into the storage vesicles and results in their immediate depletion in the spinal cord [28]. Our findings are consistent with those of a mice study that found that chronic reserpine treatment showed dose-dependent temporal blocking or delaying of PDR [20]. However, acute reserpine treatment at 1 mg/kg did not affect the convulsion patterns in mice [20]. These disparities could be attributed to the following three reasons: (1) species differences in the susceptibility to reserpine [29] with mice appearing to require higher reserpine doses for effects than fish; (2) differences in the time intervals between decapitation and reserpine administration; and (3) physiological differences between PDR and delayed convulsion even though they both involve the monoaminergic systems.

Contrastingly, pargyline treatment (30 mg/kg) significantly inhibited and prolonged delayed convulsion and latency, respectively, in red seabreams that underwent CS (Figures 1B and 2A). Pargyline is a potent nonspecific monoamine oxidase inhibitor that prevents NE metabolism; therefore, it maintains constant spinal cord NE levels [30]. The slight post-CS decrease in NE levels suggested that tonic inhibition remained (Figure 4A) and thus suppressed the rebound hyperactivity observed in the control group. Contrastingly, 40 mg/kg pargyline treatment did not affect spinal cord NE levels; moreover, it did not alter latency and duration in rats [12]. These inconsistent findings could be attributed to species differences in the NE reuptake rates. In this case, a low reuptake rate in fish could cause an increase in synaptic NE levels; contrastingly, excessive NE reuptake capacity results in mild increment without functional convulsion inhibition [31]. To further explore NE terminal involvement in delayed convulsion, future studies should employ the combination of selective adrenergic agonists/antagonists with genetically engineered approaches, including adrenoceptor knockout.

Numerous studies using pharmaceutical methods have reported the involvement of descending norepinephrinergic fibers in regulating post-decapitation convulsion [12–14]. Pre-decapitation treatment with NE-depleting agents, including LC-6-OHDA and DSP-4, resulted in rapid NE decrement by >90%, which caused a faint PDR influenced by the remaining spinal cord NE [32]. Blocking postsynaptic $\alpha_1$-adrenoceptors has been found to suppress PDR [15]. Eichbaum et al. [8] reported that post-decapitation violent clonic seizures were not associated with the central and peripheral adrenergic system given that neither adrenalectomy nor pretreatment with adrenergic receptor antagonists inhibited PDR. This suggests that noradrenergic activity in PDR occurrence is unpredictable or condition-dependent [33]. Notably, although PDR and delayed convulsions could have similar physiological responses following the sudden loss of tonic inhibition, there may be different underlying mechanisms given the numerous differences in convulsion components, including latency and duration, as well as drug responses.

Given the lack of reserpine specificity in monoamine depletion, the involvement of serotonergic or dopaminergic pathways in delayed convulsion cannot be excluded. High monoamine levels, including 5-HT, NE, and DA, could increase the thresholds for neuron activation, which depresses the excitatory postsynaptic potential evoked in neurons [20,34,35]. Furthermore, 5-HT is involved in the descending inhibitory bulbospinal pathways [36]. In our study, there were very low spinal cord DA levels even in the control group throughout the convulsion process (Figure 4B). Selective NE, but not DA, depletion has been shown to completely prevent convulsions, which indicates that PDR may be independent of DA innervation [37]. In rats, selective 5-HT depletion in the spinal cord using 5,7-dihydroxytryptamine (5,7-DHT) did not affect PDR characteristics except for latency [12]. Indeed, serotonergic or dopaminergic pathways are unlikely to be involved in delayed convulsion in fishes. However, we did not assess 5-HT levels; therefore, there is a need for further studies to investigate 5-HT involvement in delayed convulsion in fish.

Spasms or spontaneous muscle twitches are mainly caused by permanent MN excitability and the absence of inhibitory control of sensory transmission [38,39]. In rats, selective destruction of spinal monoamine terminals through intrathecal administration of 6-OHDA or 5,6-DHT significantly decreased nociceptive thresholds. This led to exaggerated spinal sensory transmission as indicated by shorter latency and more vigorous tail-flick response movements following noxious cutaneous stimulation [40]. In the bullfrog (*Lithobates catesbeianus*), there was an increase in the MN discharge rate in the spinal cord within one hour after severing at the medulla oblongata boundary; however, this increased electrophysiological activity was offset by peeling [41]. The aforementioned findings indicate the importance of descending inhibitory controls in modulating MN excitability. Given the multifactorial development of spontaneous MN activity, acute loss of NE released from descending pathways could be a crucial factor.

Delayed convulsion is characterized by self-sustained rhythmic trunk oscillation without control of the descending systems, which demonstrates that the movements have a spinal origin. Similar spontaneous rhythmic movements have been reported in certain reptiles [42], including the leopard gecko's (*Eublepharis macularius*) autotomized tail, which shows rhythmic swings and dorsoventral flips [43]. Electromyographic recordings revealed rhythmic and stereotyped motor patterns on the swinging tails, which suggests that these oscillatory movements were mediated by CPG networks [44]. We observed these spontaneous movements in fish, suggesting that delayed convulsions could involve CPG activities in response to numerous peripheral stimuli.

The SCD group showed better ATP-retaining capacity than the control group (Figure 3), which could be attributed to the inhibition of delayed convulsion. Ando et al. reported that SCD allowed retardation of ATP consumption in yellowtails and red seabreams [45]. Nakamura et al. [7] demonstrated that delaying ATP degradation and pH decrement attenuated the rigor mortis and improved the breaking strength of muscle fillets. Since delayed convulsions in red seabreams occurred at 5–30 min post-sacrifice, they could accelerate ATP decrement in muscles and cause rapid fish meat deterioration. Inhibiting monoamine

oxidase activities may effectively prevent delayed convulsions. Therefore, pharmacological manipulation of spinal cord neurotransmitter levels could be the most potent strategy for ATP preservation, which might functionally reflect the extensive inhibition of somatic and autonomic nervous system activities. Although the application of these drugs is not certified for foods yet, this study depicted the effect of modulation of neurotransmitter levels in decrease of ATP consumption. Presently, SCD destruction immediately after CS remains the mainstream approach for preventing delayed convulsions and the consequent ATP consumption in skeletal muscle given that this procedure eliminates MNs [45]. Notably, the reserpine and pargyline groups showed better retention of the ATP reserve in musculatures than the SCD group (Figure 3). This suggests that physiological activities other than delayed convulsions remained after SCD. Similarly, the superiority of neuro-modulators in preserving muscle ATP levels has been reported by a study using anesthetics, including isoeugenol [46], which is a selective monoamine oxidase A inhibitor that has been approved for aquaculture use in several countries [47,48]. Therefore, pharmacological inhibition of neural activities should be considered as an alternative practice for replacing SCD. The effects of natural monoamine inhibitors derived from herbs [49] and algae [50] are currently under investigation. Their application in fishing industry should be evaluated in the near future. In addition, we saw the huge variation in the convulsion components of fishes that were subjected to the same treatment, which might result in insignificant difference between experimental groups in this study. This limitation may be overcome by sample size enlargement.

Our findings suggest that delayed convulsions in red seabreams involve a rapid decrease in spinal cord NE levels (Figure 1). The exact mechanism underlying delayed convulsions remains unclear with respect to other involved neurotransmitters and spinal neural circuits, as well as the mechanisms underlying the species specificity. Therefore, there is a need for further research on the underlying mechanisms.

## 4. Materials and Methods

### 4.1. Fish

To determine the factors that may affect delayed convulsion, several commercial fishes in Japan were selected for this study. They reside in different environments which comprise freshwater and saltwater. Cultured purebred common carp *Cyprinus carpio* (10 fishes), tilapia *Oreochromis niloticus* (9), rainbow trout *Oncorhynchus mykiss* (10), ayu *Plecoglossus altivelis* (2), were obtained from local suppliers and maintained $\leq$ for 2 h in a 60-L water tank filled with well-aerated fresh water. Cultured purebred greenling (Ainame) *Hexagrammos otakii* (3), red seabream *Pagrus major* (35), striped jack (Shima-aji) *Caranx delicatissimus* (2), and stone flounder (Ishigarei) *Kareius bicoloratus* (2) were kept in seawater. Immediately after sacrificing through CS around the medulla oblongata, as shown in Figure 5, we measured delayed convulsion occurrence as described below. All the animal experiments conducted in this study were compliant with the institutional ethical guidance and approved by the University of Tokyo (P17-093, #AIMCB-404).

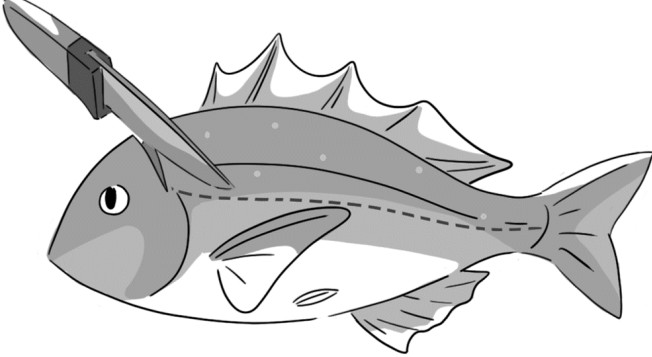

**Figure 5.** Schematic diagram of cranial spiking around medulla oblongata.

### 4.2. Measurements of Delayed Convulsion, as Well as Muscles Levels of ATP and Creatine Phosphate

Thirty-two red seabreams (720–940 g in body weight) were used to reveal the mechanism underlying delayed convulsion. This fish species was found to present stable delayed convulsion after CS. They were allocated to groups as described henceforth. Six fish were sacrificed by CS around the medulla oblongata (initial group) in order to measure the initial muscle ATP concentration in the pre-convulsive state. To assess the ATP expenses for convulsive activity, ten fish were similarly sacrificed and exposed to air at 18–20 °C until the delayed convulsion subsided (control group). As it has been speculated that delayed convulsion is implicated with abnormal neural activities in the spinal cord. Reserpine and pargyline were applied to preserve or deplete endogenous monoamine releases in the spinal cord, respectively; while, SCD was employed to completely eliminate the functions of the spinal cord. The sample sizes for the groups subjected to the following execution were downsized to prevent the experimental errors caused by the difference in time. Two different doses of reserpine were applied in this study. Four fish were intra-abdominally injected with 1 mL of 10 mg reserpine/mL dimethyl sulfoxide 2 h before being sacrificed using CS (reserpine group). Another 4 fish were injected with 1 mL of 1 mg reserpine/mL. Four fish were intra-abdominally injected with 2 mL of 30 mg pargyline hydrochloride/mL phosphate-buffered saline (PBS) 2 h before being sacrificed through CS (pargyline group). Four individuals were sacrificed by CS followed by spinal cord destruction (SCD) using a syringe air pressure extrusion unit and left at room temperature (SCD group). Each fish underwent measurement of delayed convulsion, ATP levels, and creatine phosphate levels as described below.

The head of the sacrificed fish was fixed with steel strings onto a board connected to a load cell unit (EB3200D, Shimadzu, Kyoto, Japan). Load cell signals produced by fish convulsions were recorded using an NR-2000 analog-digital converter (Keyence, Tokyo, Japan), followed by analysis on a personal computer. Estimated convulsion magnitude (gw) was obtained from the accumulation of digital signals recorded. The time span between CS and the initial signal registered was defined as latency.

Five grams of muscle strip were excised from the dorsal fast muscle of each fish immediately after death for the initial group and after convulsion for the other groups. Muscle strips were extracted using 10% perchloric acid (PCA) solution and used for ATP, creatine phosphate, and lactate analyses as previously described [51]. Briefly, 5 g dorsal muscles were homogenized in 10 mL PCA solution with a glass rod, followed by centrifugation ($900\times g$, 5 min). The resulting supernatant was removed and adjusted to pH 7 with 6 M KOH. After centrifugation ($900\times g$, 5 min), the resulting supernatant was analyzed as follows. A high performance liquid chromatography (HPLC) instrument (LC-10A HPLC system, Shimadzu, Kyoto, Japan) equipped with a UV detector and a Shenshu Pak column (Sax-1201-N, 4.6 $o \times$ 200 mm), was used in the measurement of ATP and CP. ATP was eluted in a four-step gradient with Solution A, containing 0.1 M $KH_2PO_4$ (pH 2.8) and 10% methanol and Solution B, containing 0.5 M $KH_2PO_4$ (pH 2.8) and 10% methanol at a flow rate of 1 mL/min, as follows: 100% Solution A to 2 min, 80% to 8 min, 65% to 15 min, and 20% to 25 min at 55 °C. For CP, elution was performed using Solution C, containing 20 mM $KH_2PO_4$ (pH 3.4) and 10% methanol, and Solution D, containing $KH_2PO_4$ (pH 3.4) and 10% methanol, as follows: 95% Solution C to 5 min, 90% to 10 min and 20% to 25 min at 42 °C. ATP and CP were detected at wavelengths of 254- and 210- nm, respectively. The spinal cord was excised using a syringe air pressure extrusion unit and used to assess NE and monoamines as described below.

### 4.3. Determination of Norepinephrine and Monoamines in the Spinal Cord

For another experimental set, we divided 27 cultured red seabreams (body weight: 780–830 g) into three groups as described below in order to assess the variation of monoamine and epinephrine concentrations in the spinal cords. Nine fish in the reserpine group were intra-abdominally injected with 1 mL of 10 mg reserpine/mL dimethyl sulfoxide 2 h before being sacrificed through CS. Nine fish in the pargyline group were intra-abdominally

injected with 2 mL of 30 mg pargyline hydrochloride/mL PBS 2 h before CS. Finally, 9 fish in the control group were intra-abdominally injected with 2 mL of PBS 2 h before CS. The spinal cord was excised immediately ($n = 3$), 5 min ($n = 3$), and 20 min ($n = 3$) after sacrifice for each group using a syringe air pressure unit. The sampling time points were determined by the onset time of the delayed convulsion that ranging from 5~20 min.

The excised spinal cord was immediately homogenized in five volumes of 0.1 M HCl at 8000 rpm for 30 s using an Ultra-Turrax T25 (IKA Asia, Kuala Lumpur, Malaysia). Next, the homogenates were centrifuged at $5000 \times g$ for 20 min, followed by analysis of the yielded supernatants in an HPLC system at an injection volume of 20 μL. HPLC analysis was performed using a column ODP-50G Asahipak (Showa Denko, Tokyo, Japan) and the LC-10A HPLC system (Shimadzu, Kyoto, Japan) equipped with a fluorescent detector. Regarding the mobile phase, we added 10% ($v/v$) methanol into the solution (pH 3.0) containing 35 mM citric acid, 12.5 mM $Na_2HPO_4 \cdot 12H_2O$, 0.05 mM ethylenediamine disodium salt dihydrate, and 300 mg/L 1-heptanesulfonic acid sodium salt. Moreover, we employed a flow rate of 0.6 mL/min at an ambient temperature. NE and monoamines were separately derivatized using solution containing 0.3 M $Na_2CO_3$, 0.086 M $K_2SO_4$, 0.18 M $H_3BO_3$, 800 mg/L o-phthalic acid, 0.14% ($v/v$) ethanol, 0.2% ($v/v$) 2-mercaptoethanol, 0.04% ($v/v$) Briji-35, 0.1 M $Na_2CO_3$, 0.029 M $K_2SO_4$, 0.06 M $H_3BO_3$, and 0.04% NaClO solution (Nacalai, Kyoto, Japan) at 39 °C. Derivatized NE and monoamines were detected using a fluorescent detector at a 350- and 450- nm excitation and emission wavelengths, respectively.

### 4.4. Statistical Analyses

All statistical analyses were performed using Sigmaplot v14.0 (Systat Software, Inc., San Jose, CA, USA). Data were presented as group means ± standardized deviation. In Figure 2, the variables of the experimental groups were compared with the value of their corresponding control group for individual convulsion components. In Figure 4, the values of experimental groups were compared with the control group for each time point. All statistical comparisons were evaluated using Dunnett's test after Bartlett's test. Statistical significance was set as $p < 0.05$. In Figure 3, the relationship between the remaining ATP equivalent levels and the convulsion magnitudes were evaluated through linear regression analysis.

**Author Contributions:** H.U. designed and performed the experiments; C.-L.L. and H.U. analyzed the data.; C.-L.L., Y.K. and H.U. wrote the manuscript; H.U. contributed to the fund acquisition. All authors have read and agreed to the published version of the manuscript.

**Funding:** This research was funded by the Japan Society for the Promotion of Science.

**Institutional Review Board Statement:** All the animal experiments conducted in this study were compliant with the institutional ethical guidance and approved by the University of Tokyo (P17-093, #AIMCB-404).

**Conflicts of Interest:** The authors declare no conflict of interest.

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
