# Peer review of "Mechanism of Delayed Convulsion in Fish: The Actions of Norepinephrine in Spinal Cord"

_fishes, doi:10.3390/fishes6020012_

Round 1

Reviewer 1 Report

Dear Authors,

Let me congratulate you on this original study, which brings new knowledge to the physiological process behind the possible mechanism responsible for delayed convulsion in fish killed by cranial spiking.

However, you have several problems that need to be addressed in the manuscript:

1. Introduction

Overall the text has several ideas that miss the connection between them and can make this part of the manuscript hard to read. Please rewrite this introduction to improve the readability. You have several suggestions for the annexed PDF.

2. Results

2.1 Among-species differences in delayed convulsion measures

In this part, you have several problems with the experimental design, mainly regarding the number of individuals analysed per species. That bring the following questions:

- First of all, why did you state that the fish are from different environments when the main difference is in the salinity? After all, in saltwater fish, you also have different environments between species...

Why did you only use two ou three individuals on delayed convulsion analysis in greenling, ayu, striped jack, and plaice? Don't you think that the low number of analysed fish on this species introduces a bias on the results?

Please see other corrections and suggestion on the annexed PDF.

2.2. Effects of pharmacological compounds involving monoamine metabolism on delayed convulsion in red seabream

Please see corrections and suggestion on the annexed PDF.

2.3. Relationship between delayed convulsion and changes in ATP and creatine phosphate levels

Please see corrections and suggestion on the annexed PDF.

2.4. Changes in spinal cord norepinephrine and monoamine levels

Please see corrections and suggestion on the annexed PDF.

3. Discussion

Overall this part of the manuscript is difficult to read. Please rewrite some parts of the text to improve the readability. You have several suggestions for the annexed PDF. I would also suggest that you insert a figure with a probable mechanism involved in delayed convulsions in red sea bream, which will improve vastly the readability of this manuscript part and also the readers' interest.

I also have a question that I hope that you can address:

- On line 171 of the manuscript, you state that "The red seabream specifically exhibited a representative delayed convulsion.". What do you consider a representative delayed convulsion?

4. Materials and Methods

This part of the manuscript is well written and clear to the reader. However, I have some issues with part 4.1 and 4.2 that should be addressed.

4.1. Fish

Please insert the number of fish used in this experiment, since this information is present in the Results section.

4.2. Measurements of delayed convulsion, as well as muscles levels of ATP and creatine phosphate

Please describe briefly the methodologies used for ATP, creatine phosphate and lactate in this part, besides the bibliographic reference.

References

Please correct the issues detected in this section (see annexed PDF).

Best regards,

Author Response

Responses to Reviewer 1

Thank you for your helpful comments. We were attempting to correct the manuscript with our utmost efforts.

  1. Introduction

Overall the text has several ideas that miss the connection between them and can make this part of the manuscript hard to read. Please rewrite this introduction to improve the readability. You have several suggestions for the annexed PDF.

This transition of sashimi to techniques of fish sacrifice is awkward. Please rephrase in order to include why the sacrifice methods are important for freshness.

We have revised the introduction according to the comments. Please see Line 30-33. 

I miss here a closing part and a transition to the next part of the texts, like saying that the physiological process behind the convulsion in unknown, for example. Please rephase this part.

We have rephased the texts to make a better transition. Please see Line 38-42.

This paragraph is confusing, since if has parts that belong to Material and Methods ("Based on our results, we selected (...) with neurotransmitter contents" - This part should be on Material and Methods, 4.2. Measurements of delayed convulsion, as well as muscles levels of ATP and creatine phosphate) or Conclusions ("The findings of this (...) on fish flesh quality").

Please rephrase this part, making clear to the reader what is the scientific hypothesis and the objective of your work.

We have deleted improper parts of this paragraph and rephrased to make the scientific hypothesis and the objective of our work clearer. Please see Line 62-65.

  1. Results

2.1 Among-species differences in delayed convulsion measures

In this part, you have several problems with the experimental design, mainly regarding the number of individuals analyzed per species. That bring the following questions:

First of all, why did you state that the fish are from different environments when the main difference is in the salinity? After all, in saltwater fish, you also have different environments between species...

We have rephrased the statement and specified that the fish were cultivated in freshwater or saltwater environment. Please see Line 68-69.

Why did you only use two or three individuals on delayed convulsion analysis in greenling, ayu, striped jack, and plaice? Don't you think that the low number of analyzed fish on this species introduces a bias on the results?

This is an overview work without statistical analysis, because these 4 species are difficult to be kept in a laboratory water tank system. There might be bias in the results, but we can overview species-specific profiles of delayed convulsion.

Please insert species scientific name, as you mention them for the first time in the manuscript.

We have inserted the species scientific names as your instruction. Please see Line 69-72.

The Table legend should not include a result description. Please rephrase.

We have deleted the result description.

I suggest that the table present also the different environments of the fish (freshwater, ...). This would improve the table and provide a very useful information to the manuscript readers.

We recategorized those fish into freshwater and saltwater group in Table 1. Although natural ayu and rainbow trout swim between the river and the sea, the fish used in this study were cultured in freshwater system and had never had access to seawater.

2.2. Effects of pharmacological compounds involving monoamine metabolism on delayed convulsion in red seabream

Please see corrections and suggestion on the annexed PDF.

We are sorry but we did not see any indication of corrections and suggestion in the PDF.

2.3. Relationship between delayed convulsion and changes in ATP and creatine phosphate levels

Please see corrections and suggestion on the annexed PDF.

This is the Figure 2 analysis. Please rephrase to a proper legend for Figure 2.

We have deleted the unnecessary description in Figure 2 legend.

2.4. Changes in spinal cord norepinephrine and monoamine levels

Please see corrections and suggestion on the annexed PDF.

Please separate this part of the text from the Figure 3 legend.

We have deleted the unnecessary description in Figure 3 legend.

  1. Discussion

Overall this part of the manuscript is difficult to read. Please rewrite some parts of the text to improve the readability. You have several suggestions for the annexed PDF.

I would also suggest that you insert a figure with a probable mechanism involved in delayed convulsions in red sea bream, which will improve vastly the readability of this manuscript part and also the readers’ interest.

Thank you for the suggestion. We have inserted an illustration to present the slaughter procedure to make the readers quickly understand. Please see Figure 5.

I also have a question that I hope that you can address:

On line 171 of the manuscript, you state that "The red seabream specifically exhibited a representative delayed convulsion.". What do you consider a representative delayed convulsion?

We think a representative delayed convulsion means a moderate period of time for latency and convulsion duration. Both too short or too lengthy increase the difficulty and labor efforts for the experiments. We have put additional description for this issue. Please see Line 164-165.  

The authors need to start the discussion in a different way. Please rephrase this sentence to improve the readability of the manuscript.

We have revised it in improving the readability. Please see Line 164-165.

Please eliminate this part of the text.

We have eliminated. Please see Line 160.

Please rephrase this paragraph. This is very confusing to the reader.

We have deleted this paragraph.

Please eliminate this.

We have eliminated. Please see Line 163.

Please eliminate this.

We have eliminated. Please see Line 165.

Please consider rephrasing to "Contrastingly, the freshwater fish species

carp and tilapia did not present convulsive responses."

We have replaced this sentence as your suggestion and made little modification. Please see Line 166-168.

Please eliminate this.

We have eliminated. Please see Line 179.

Please insert bibliographic reference.

We have inserted the bibliographic reference. Please see Line 183.

  1. Materials and Methods

This part of the manuscript is well written and clear to the reader. However, I have some issues with part 4.1 and 4.2 that should be addressed.

4.1. Fish

Please insert the number of fish used in this experiment, since this information is present in the Results section.

We have inserted the number for each fish species after its name. Please see Line 304-309.

4.2. Measurements of delayed convulsion, as well as muscles levels of ATP and creatine phosphate

Please describe briefly the methodologies used for ATP, creatine phosphate and lactate in this part, besides the bibliographic reference.

We have briefly described the methodologies and procedures for ATP and creatine phosphate detection. Please see Line 347-361.

References

Please correct the issues detected in this section (see annexed PDF).

We have corrected the incorrect reference format as your instructions.

Reviewer 2 Report

Among the numerous techniques currently adopted for the sacrifice of fish, the Cranial spiking (CS) is widely used as slaughtering method, useful for delaying the rigor mortis progress of fish muscle. Cranial spiking consists in driving a sharp spike into the hindbrain region of the fish causing immediate unconsciousness of the animal. However, delayed convulsive phenomena affecting meat quality (flavour and texture) are common consequences of CS. Cheng-Linn Lee and colleagues presented a study aimed to clarify the mechanism involved in the delayed convulsion and investigate its influence on ATP consumption. Firstly, several fish species were evaluated for their convulsion profiles after CS, in particular: carp, tilapia, rainbow trout, ayu, greenling, striped jack, plaice and red sea bream. Among tested fish, red seabream exhibited representative delayed convulsions so further pharmacological analyses aimed to investigate the association of delayed convulsion with neurotransmitter (norepinephrine) contents were carried out on this species. Red seabreams were treated with a monoamine-depleting agent, reserpine, or with a monoamine oxidase inhibitor, pargyline, two hours before CS. Furthermore, Spinal cord destruction (SCD) was employed to completely prevent spinal cord functions of the fish in another fish group. The treatment of reserpine, pargyline or SCD showed a significant inhibition of delayed convulsion responses compared to the control group (only CS treatment). In light of presented results authors suggest that delayed convulsion may be the main ATP-consuming response. In addition, Cheng-Linn Lee and colleagues state that that delayed clonic convulsions in red seabreams are related to the rapid decrease in spinal cord of norepinephrine levels, which activate the rebound motor neuron hyperactivity. In my opinion this manuscript reports a relevant and rigorous study supported by an appropriate experimental design. I recommend publication of the manuscript, with only a few minor corrections. Please find more specific comments below.

Line 46 please add the reference number of this citation “Fukuda et al.”.

Line 158 please review this title grammatically.

Line 189 please add the reference number of this citation “Roberts et al.”.

Author Response

Responses to Reviewer 2

Thank you for your helpful comments. We were attempting to correct the manuscript with our utmost efforts.

Line 46 please add the reference number of this citation “Fukuda et al.”.

We have added the reference number after “Fukuda et al.”. Please see Line 49.

Line 158 please review this title grammatically.

We have shortened the title and corrected the grammatical mistakes. Please see Line 156.

Line 189 please add the reference number of this citation “Roberts et al.”.

We have added reference number after “Roberts et al.”. Please see Line 183.

Reviewer 3 Report

The manuscript fishes-1144687 entailed “Mechanism of delayed convulsion in fish: The actions of norepinephrine in spinal cord” aimed to elucidate the neural mechanism underlying delayed convulsion in fishes. In particular, authors hypothesized that post-cranial spiking delayed convulsion in fish is associated with norepinephrine depletion in the spinal cord. To do this, 27 red seabreams were treated with reserpine (a monoamine-depleting agent), or with pargyline (a monoamine oxidase inhibitor) two hours before cranial spiking.

The MS provides new and interesting data. The introduction, although brief, define the current state of the research field, the purpose of the work and its significance, including specific hypotheses being tested. However, my main concern regards the statistical analysis. The section 4.4. need to be improved. Did your data normally distributed? Why did you choose to apply the Dunnett's test? Also, section “2.3. Relationship between delayed convulsion and changes in ATP and creatine phosphate levels” need to be revised and improved. Appropriate statistical analysis need to be perform to demonstrate the correlation. The statement in the figure 2 “The scatter plot demonstrated the correlation between the magnitudes of delayed convulsion and remaining ATP + creatine phosphate (CP) contents in the dorsal skeletal muscles at post-convulsion state” is not demonstrated. Did you perform a correlation test or a regression analysis? Please improve the results accordingly.

Minor comments

Line 175. Authors state that “… findings suggest that the convulsion occurrence and intensity could be related to the musculature salt contents” but no data were reported. Did you have some information/data about it? Please, improve the discussion accordingly.

References need to be checked and revised following the authors guidelines.

Author Response

Responses to Reviewer 3

Thank you for your helpful comments. We were attempting to correct the manuscript with our utmost efforts.

The section 4.4. need to be improved. Did your data normally distributed? Why did you choose to apply the Dunnett's test?

We did Bartlett’s test first, and then Dunnett's test. Because we want to compare the data between control and others, we used Dunnett’s test. We inserted Bartlett’s test in the text.

Also, section “2.3. Relationship between delayed convulsion and changes in ATP and creatine phosphate levels” need to be revised and improved. Appropriate statistical analysis need to be perform to demonstrate the correlation.

We performed Pearson correlation and obtained the correlation coefficient (r), - 0.8013, and R square, 0.6422, suggesting a negative correlation between delayed convulsion magnitude and the remaining ATP and creatine phosphate levels.

The statement in the figure 2 “The scatter plot demonstrated the correlation between the magnitudes of delayed convulsion and remaining ATP + creatine phosphate (CP) contents in the dorsal skeletal muscles at post-convulsion state” is not demonstrated. Did you perform a correlation test or a regression analysis? Please improve the results accordingly.

We did perform regression analysis. We apologize for the incomplete data presentation. The values of regression analysis have been added to the figure.

Line 175. Authors state that “… findings suggest that the convulsion occurrence and intensity could be related to the musculature salt contents” but no data were reported. Did you have some information/data about it? Please, improve the discussion accordingly.

We cannot find adequate information from literatures to support this theory, thus we rephrased this paragraph. Please see Line 168-172.

References need to be checked and revised following the authors guidelines.

We have corrected the references following the authors guidelines.

Round 2

Reviewer 1 Report

Dear Authors,

Let me congratulate you again on this original and innovative study, and also on the massive improvements that you had performed in this manuscript. They greatly improved the quality and readability of your work.

Your answer to my questions was also satisfactory and had enlightened some doubts that I had about your manuscript.

I only have detected some minor issues in the manuscript like the absence of a reference to Figure 1 in the manuscript and a missing reference on the text, for example, that you can check on the suggestions placed on the annexed PDF.

Best regards,

Author Response

Dear reviewer,

Thank you again for revising our manuscript and giving additional comments. We believe that these comments have made this manuscript reach a higher scientific level.

Dear Authors,

Let me congratulate you again on this original and innovative study, and also on the massive improvements that you had performed in this manuscript. They greatly improved the quality and readability of your work.

Your answer to my questions was also satisfactory and had enlightened some doubts that I had about your manuscript.

I only have detected some minor issues in the manuscript like the absence of a reference to Figure 1 in the manuscript and a missing reference on the text, for example, that you can check on the suggestions placed on the annexed PDF.

Best regards,

Please insert a reference to Figure 1 in the manuscript.

We have inserted the reference number. Please see Line 63.

Please eliminate one of the final marking points.

We have eliminated it. Please see Line 142.

You must complete the Figure legend with this "Fish subjected to different treatments are represented as: â—‹137 (control), â–¼(1mg reserpine), â–³(10 mg reserpine), ï‚¢ (pargyline) and ï‚£ (spinal cord 138 destruction). Statistically significant difference is shown as * 120 p < 0.05, ** p < 0.01. Not significant, n.s. "

We have added the related information to the legend of figure 4. Please see Line 161.

Please replace by "with the limited information available".

Thank you for the sentence revision. We have replaced it. Please see Line 177.

Please insert the reference number.

We have inserted the reference number. Please see Line 274.

Reviewer 3 Report

The authors addressed all my comments. However, I think that they should modify the figure 3 legend. They showed the linear regression analysis. Correlation quantifies the direction and strength of the relationship between two numeric variables, X and Y, and always lies between -1.0 and 1.0. Simple linear regression relates X to Y through an equation of the form Y = a + bX. 

Also, figures quality need to be improved before publication.

Author Response

Dear reviewer

Thank you again for revising our manuscript and giving additional comments. We believe that these comments have made this manuscript reach a higher scientific level.

The authors addressed all my comments. However, I think that they should modify the figure 3 legend. They showed the linear regression analysis. Correlation quantifies the direction and strength of the relationship between two numeric variables, X and Y, and always lies between -1.0 and 1.0. Simple linear regression relates X to Y through an equation of the form Y = a + bX.

We have added “Simple linear regression relates X to Y through an equation of Y = 14.28 – 0.0245*X.” to the legend of figure 3. We also reduced the significant figure of y0 of regression equation. Please see Line 141.

Also, figures quality need to be improved before publication.

We have improved the figure quality of figure 3 according to your instruction.